# Electrical and Thermal Conductivities of Single Cu_x_O Nanowires

**DOI:** 10.3390/nano13212822

**Published:** 2023-10-25

**Authors:** Ivan De Carlo, Luisa Baudino, Petr Klapetek, Mara Serrapede, Fabio Michieletti, Natascia De Leo, Fabrizio Pirri, Luca Boarino, Andrea Lamberti, Gianluca Milano

**Affiliations:** 1Advanced Materials Metrology and Life Sciences Division, Istituto Nazionale di Ricerca Metrologica (INRiM), 10135 Turin, Italy; ivan.decarlo@polito.it (I.D.C.); fabio.michieletti@polito.it (F.M.); n.deleo@inrim.it (N.D.L.); l.boarino@inrim.it (L.B.); 2Department of Electronics and Telecommunications, Politecnico di Torino, 10129 Turin, Italy; 3Department of Applied Science and Technology, Politecnico di Torino, 10129 Turin, Italy; luisa.baudino@polito.it (L.B.); mara.serrapede@polito.it (M.S.); andrea.lamberti@polito.it (A.L.); 4Czech Metrology Institute, Okružní 31, 638 00 Brno, Czech Republic; pklapetek@cmi.cz; 5Center for Sustainable Future Technologies @Polito, Istituto Italiano di Tecnologia (IIT), 10144 Turin, Italy

**Keywords:** CuO, nanowires, single-nanowire electrical characterization, SThM on single nanowire

## Abstract

Copper oxide nanowires (NWs) are promising elements for the realization of a wide range of devices for low-power electronics, gas sensors, and energy storage applications, due to their high aspect ratio, low environmental impact, and cost-effective manufacturing. Here, we report on the electrical and thermal properties of copper oxide NWs synthetized through thermal growth directly on copper foil. Structural characterization revealed that the growth process resulted in the formation of vertically aligned NWs on the Cu growth substrate, while the investigation of chemical composition revealed that the NWs were composed of CuO rather than Cu_2_O. The electrical characterization of single-NW-based devices, in which single NWs were contacted by Cu electrodes, revealed that the NWs were characterized by a conductivity of 7.6 × 10^−2^ S∙cm^−1^. The effect of the metal–insulator interface at the NW–electrode contact was analyzed by comparing characterizations in two-terminal and four-terminal configurations. The effective thermal conductivity of single CuO NWs placed on a substrate was measured using Scanning Thermal Microscopy (SThM), providing a value of 2.6 W∙m^−1^∙K^−1^, and using a simple Finite Difference model, an estimate for the thermal conductivity of the nanowire itself was obtained as 3.1 W∙m^−1^∙K^−1^. By shedding new light on the electrical and thermal properties of single CuO NWs, these results can be exploited for the rational design of a wide range of optoelectronic devices based on NWs.

## 1. Introduction

Semiconductor nanowires (NWs) grown with a bottom–up approach represent unique nanoscale building blocks for the scaling of semiconductor electronics as well as suitable platforms for the fundamental understanding of nanoscale physical phenomena [1,2]. Offering unique properties such as a high surface-to-volume ratio and tunable charge carrier transport, these quasi-one-dimensional structures find applications in a wide range of technological solutions, including high-speed and low-power electronics, photovoltaic devices, electrochemical energy storage devices, and thermoelectric applications [3,4,5,6,7,8,9,10]. Cupric oxide exhibits the characteristics of a p-type semiconductor as a result of the existence of copper vacancies with negative charges and interstitial oxygen [11,12]. Its bandgap energy is relatively narrow, falling within the range of 1.3 to 1.7 eV, according to reported fundings [13,14]. In this context, CuO and Cu_2_O nanowires (NWs) synthetized through thermal growth directly on copper foil hold significant potential for various applications in electronic devices and sensors [15,16,17,18]. In electronics, CuO NWs can be used to fabricate high-performance field-effect transistors (FETs), nanoscale interconnects, and optoelectronic devices [17,19,20,21]. Their compatibility with copper foil enables large-scale, low-cost manufacturing processes, making them highly desirable for flexible and printable electronics applications [22]. In the field of sensors, CuO and Cu_2_O NWs exhibit excellent gas-sensing properties [16,23,24]. These NWs can detect a wide range of gases, including toxic gases, volatile organic compounds (VOCs), and environmental pollutants [18,25,26]. Their high sensitivity, selectivity, and fast response time make them ideal for gas sensors in industrial and environmental monitoring applications. Additionally, CuO and Cu_2_O NWs on copper foil have been explored for energy storage applications, such as lithium-ion batteries and supercapacitors, due to their large surface area and good electrochemical performance [27,28]. Polycrystalline CuO NWs are also promising for resistive switching applications, where the presence of different oxide phases and high density of defects can locally enhance the electric field, facilitating the oxygen vacancies’ diffusion through the nanostructure [29,30,31]. In the field of optoelectronics, CuO NWs have demonstrated potential applications in photodetectors, photovoltaics, and photocatalysis due to their p-type semiconducting behavior, unique bandgap tunability, light absorption properties, and their photocatalytic activity when coupled with a noble metal or coated with other metal-oxides for the realization of p–n heterojunctions based on semiconducting NWs [32,33,34,35,36,37,38]. Overall, the thermal growth of CuO or Cu_2_O NWs directly on copper foil offers potential for advancing electronic devices and sensors, paving the way for high-performance, cost-effective, and environmentally friendly technological solutions [28]. In this paper, we report on the electrical and thermal conductivities of Cu_x_O NWs obtained by the thermal oxidation of Cu foil. X-ray diffraction (XRD) revealed the presence of two copper oxide phases, cuprite (Cu_2_O) and tenorite (CuO), and information on the NWs’ growth. X-ray photoelectron spectroscopy (XPS) revealed the composition of the NWs, which proved to be CuO rather than Cu_2_O. Investigation on electrical conductivity of single NWs was carried out, revealing an increase in the conduction in vacuum, and a conductivity of 7.6 × 10^−2^ S∙cm^−1^. The effective thermal conductivity of single CuO NWs was measured using Scanning Thermal Microscopy (SThM), and by means of a Finite Difference method, the thermal conductivity of the NW itself was evaluated, yielding the values 2.6 W∙m^−1^∙K^−1^ and 3.1 W∙m^−1^∙K^−1^, respectively. The results obtained for the electrical and thermal properties may be exploited for the implementation of CuO NWs in optoelectronic devices based on single NWs.

## 2. Materials and Methods

### 2.1. Growth of Cu_x_O NWs

The Cu_x_O NWs were grown through a bottom–up approach based on the thermal annealing of a copper foil in ambient conditions, exploiting the oxide film’s evolution and the NWs’ subsequent growth [28]. The Cu foil (30 mm thick, 99.9% purity), cut into 20 cm^2^ samples, was cleaned in an acetone and ethanol ultrasonic bath and then dried under N_2_ flow. The removal of the native oxide layer was performed through a bath in 1 M HCl aqueous solution for 2 min, followed by a bath in deionized water, and then dried under N_2_ flow. The Cu samples were finally annealed in an ambient air atmosphere, at 430 °C, in a convection oven, leading to the copper foil’s thermal oxidation and the Cu_x_O NWs’ growth.

### 2.2. Morphological, Structural, and Chemical Characterization

The morphology of the samples obtained by thermal oxidation was investigated through Field Emission Scanning Electron Microscopy (FESEM), using an FEI Inspect F system (Field Electron and Ion Company, Hillsboro, OR, USA), at an applied voltage of 10 kV and a working distance of 10 mm, varying the magnification from 2000 times to 50,000 times depending on the features’ size.

A Panalytical Empyrean powder diffractometer equipped with a 1D PIXcel detector was used for the XRD analysis (Malvern PANalytical, Malvern, UK) of the sample. The diffractograms were collected in the Bragg–Brentano reflection mode by using Cu Kα_1/2_ radiation, at an operating voltage of 40 kV and a tube current of 30 mA. XRD data were recorded at 20 °C, 4.6 bar, and 20% humidity. The sample was placed onto a “zero-background” holder thermalized at room temperature. The instrumental broadening was computed using the Caglioti equation based on the reflections of a standard LaB_6_ powder NIST660a. The measurements were carried out in continuous mode with a step size of 2 θ = 0.0131° and a data time per step of 1500 s. QualX equipped with the COD database was employed for qualitative phase determination, and the MAUD free software was used for quantitative analysis and refinement. By recording multiple reflections at different times, no variations on the sample were observed in 1 day of experimentation, suggesting the stability of the NWs at such room conditions and exposure doses.

The chemical composition of the Cu_x_O NW matrix samples was investigated through XPS using a PHI 5000 VersaProbe system (Physical Electronics, Inc. (PHI), Chanhassen, MN, USA). Monochromatic Al Kα (1486.6 eV) was used as an X-ray source, and charge compensation during the measurements was accomplished with a combined electron and Ar^+^ neutralizer system. Wide-energy and high-resolution (HR) XPS spectra were collected in ultra-high vacuum (2 × 10^−10^ mbar) after leaving samples degassing overnight. The analyses were repeated on three different areas of the sample to ensure compositional homogeneity and processed using CasaXPS software (version 2.3.18). HR spectra deconvolution into individual mixed Gaussian–Lorentzian peaks was obtained after Shirley background subtraction, and the binding energy (BE) was calibrated with respect to the C 1s position for adventitious carbon (284.5 eV).

### 2.3. Single-NW-Based Device Fabrication

Single Cu_x_O NW–based devices were realized by means of a combination of laser lithography, metal evaporation, and electron beam lithography (EBL), following the procedure adopted in previous works [39,40,41]. For this purpose, a commercial silicon substrate with 1 μm of thermal oxide was used to electrically characterize the NWs. A micrometric mask of Ti/Au (5 nm, 200 nm) was then designed through Heidelberg µpg101 (Heidelberg Instruments Mikrotechnik GmbH, Heidelberg, Germany), a laser lithography system, and RF sputtering in Ar plasma (around 5 × 10^−3^ mbar). The Cu_x_O NWs were transferred from the original substrate to the central part of the pre-patterned substrate, with the help of a scratching tool, near the Ti/Au tracks [41]. The NWs were randomly spread on the new substrate; therefore, their exact position was determined through FESEM. The suitable NWs were selected, and an ad hoc design for the electrical contacts between the NWs and the micrometric tracks was designed. An FEI Quanta 3D (Field Electron and Ion Company, Hillsboro, OR, USA) equipped with a Nabity NPGS pattern generator (JC Nabity Lithography Systems, Bozeman, MT, USA) was employed to define the second mask, over a PMMA A4 layer (300 nm thick), and then, 50 nm of Cu was deposited through e-beam evaporation. The excess metal was finally removed by lift-off in an acetone bath. To avoid degradation of the metal-oxide nanostructure when exposed to water, water was not used during the fabrication process.

### 2.4. Electrical Characterization

The electrical characterization of the device was performed through a Keithley 6430 (Keithley Instruments, Solon, OH, USA) connected to a vacuum chamber with a Leybold (Leybold GmbH, Cologne, Germany) pumping system attached to it. The single Cu_x_O NWs were wire-bonded to a sample holder, which was previously short-circuited in order to prevent any NWs’ breakdown due to electrical discharge during the wire-bonding. The sample holder was then screwed inside the vacuum chamber. This configuration allowed us to perform both 2-wire and 4-wire characteristics and to extrapolate the electrical conductivity of the material. At first, the characterization of the 2-terminal device was performed; to understand the junction behavior and the contact resistance, a voltage ramp was performed, first in ambient pressure and then in vacuum, at the base pressure of 2 × 10^−3^ mbar, allowing a comparison of the electrical response depending on the ambient conditions. After that, the 4-terminal characteristic of the Cu_x_O NW was measured, to understand the NW’s electrical conductivity.

### 2.5. Thermal Characterization

The thermal conductivity of single NWs was measured using SThM on a Dimension Icon Scanning Probe Microscope (Bruker Corporation, Billerica, MA, USA) with VITA SThM electronics and VITA-DM-GLA probes (Anasys Instruments Corp, Santa Barbara, CA, USA). The Cu_x_O NWs were spread on the pre-patterned substrate, similarly to the electrical contact fabrication procedure, and after that, through FESEM characterization, the exact position of the NWs was determined. SThM probes were calibrated using a set of bulk material samples measured by the French national metrology institute LNE in the framework of the FP7 project Quantiheat using the laser flash method. Moreover, to relate the SThM measurements to the numerical model, silicon dioxide steps on silicon were used [42], with known effective thermal conductivities taken from [42]. Measurements were performed in Force Volume mode to prevent movement of NWs over the surface, as observed in contact mode. Moreover, this provided better control over the contact force, which was set to 90 nN. The measured data were processed using Gwyddion open-source software [43]. In all measurements, including calibration, the SThM signal was referenced to a value far from the sample to prevent the impact of electronic drifts. Probe geometry was estimated using blind tip estimation on a rough silicon sample, providing an average probe radius of 60 nm, and the NW diameter was determined from tapping-mode Atomic Force Microscopy measurements. To model the heat power flowing through the NW, the electronics settings and environmental factors were assumed to be constant during the experiment, apart from the probe–sample area geometry changes. To provide a coarse estimate of the uncertainty related to the numerical model, simulations were performed with different probe radii, different contact resistances, and different substrate conductivities. The whole measurement and data interpretation procedure was performed twice, with two different SThM probes and on the same NWs.

## 3. Results

### 3.1. Morphological Characterization

The morphology of the NW array composed of vertically aligned Cu_x_O NWs obtained via the bottom–up thermal oxidation on Cu foil was investigated using FESEM. Figure 1 reports the low- (Figure 1a) and high-magnification (Figure 1b,c) images of the Cu_x_O NW matrix. Besides revealing the presence of Cu_x_O NWs at the surface of the copper foil, the low-magnification image highlights the uniformity and homogeneity of the NWs’ growth over a large scale, while the high-magnification images show the NW forest and their vertical alignment. In particular, SEM imaging revealed that the Cu_x_O NWs were characterized by a diameter of ~60 nm.

### 3.2. Structural Properties

The crystal structure of the Cu_x_O NW sample matrix was investigated through XRD to determine the distinct phases of the sample and/or the presence of amorphous components. The diffractogram (see Figure 2) demonstrated the presence of two distinct oxide reflections on the copper foil surface after the thermal treatments: a bottom layer consisting of cuprite (COD 9007497) and a top layer consisting of tenorite with at least two different crystalline sizes (COD 9016105 and 9014580). The main parameters retrieved by the Rietveld refinement for the three crystal structures are reported in Appendix A. It is worth noticing that the cuprite and one of the tenorite phases had quite similar sizes, while the second phase of tenorite had a far larger crystal of 2530 Å on average. Cuprite represented 44% of the sample, while the remaining 56% was tenorite. Interestingly, when the two phases of tenorite were computed, metallic copper disappeared from the refinement. When copper foil is exposed to high temperatures in ambient atmosphere, the surface undergoes oxidation. This process first leads to the formation of a Cu_2_O layer, followed by the subsequent development of a layer consisting of CuO on the top of Cu_2_O [44]. The growth of CuO NWs relies on copper diffusion from the substrate foil through both the oxide layers via stress induced at the interface of these layers [45,46,47,48]. This mechanism is confirmed by the presence of both Cu_x_O phases, cuprite and tenorite, which are a direct consequence of the Cu foil’s thermal oxidation: on the copper surface, there are two oxide layers, the bottom composed by Cu_2_O and the top composed by CuO [28,44,49,50].

### 3.3. Chemical Properties

The chemical composition of the samples was studied by means of XPS, which can determine the copper oxidation state by analyzing the Cu 2p and Cu LMM regions. As previously mentioned, the spectra were acquired in three different areas of the sample. However, since the spectra were almost superimposable, thus showing excellent homogeneity, a representative spectrum is here shown and the average value for the Auger parameter calculations is reported. By looking at the Cu 2p region (see Figure 3a), one can easily see that the copper ions were present as Cu^2+^, and thus the resulting oxide was mainly CuO. This fact is confirmed by the presence of strong satellites at both 938–946 eV and 958–966 eV, whereas Cu_2_O samples only have a weak satellite at 942–948 eV. Furthermore, a broad Cu 2p_3/2_ peak was present at 933 eV, whereas, in Cu_2_O samples, this peak is slimmer and at higher energies [28,51,52]. Further confirmation was acquired by analyzing the Cu Auger peak (see Figure 3b). Indeed, the peak appeared at 917.7 eV and the modified Auger parameter results were 1851.49 eV, perfectly in line with the literature values of (1851.5 ± 0.4) eV [53,54]. Finally, the O 1s HR spectrum could be deconvoluted into two components (see Figure 3c), respectively, the lattice oxide (at 529.8 eV) and the defective oxide at (531.4 eV). The lattice component, amounting to almost two-thirds of the oxygen amount, was slightly lower than the expected values for this kind of sample but still fell within the expected ranges [54,55]. The survey scan and the deconvolution of the adventitious carbon peak for calibration purposes are available in Appendix A, respectively. The presence of CuO as the resulting oxide from the XPS spectra is in line with the growth model previously investigated by several research groups [11,49,50,56,57]. The CuO NWs grow on a CuO top oxide layer that forms over a Cu_2_O bottom layer.

### 3.4. Electrical Properties

Figure 4a reports an SEM image of a single-NW-based device. As can be observed, the single NW was contacted by means of four electrodes, allowing characterization of the nanostructure in both two-terminal and four-terminal configurations. First, the behavior of the Cu/Cu_x_O junction was investigated by applying a voltage ramp (range +10 V/−10 V) in between two consecutive electrodes. The measured current showed symmetric and nonlinear I–V curves, which denoted that the conduction mechanism through the junction was not ohmic. This is because the single-NW device in two-terminal configuration can be modeled as two back-to-back Schottky diodes in series with the NW resistance, as previously discussed in [58,59,60]. In these circumstances, one of the two diodes is inversely polarized when a voltage bias is applied, and therefore, the current flowing through the NW has a nontrivial behavior [58,61,62]. Figure 4b shows the I–V curve of the NW-based device at ambient pressure and in vacuum (2 × 10^−3^ mbar). As it is possible to observe, the I–V characteristic is slightly influenced by environmental conditions, with a more conductive behavior that can be observed by decreasing the ambient pressure.

This phenomenon could be related to the adsorbed species, which, in this case, led to a reduction in the NWs’ conductivity [23,39,63,64]. When metal oxide NWs are exposed to air, oxygen species are adsorbed due to the interaction of oxygen molecules with crystal defects sites present on the NW surface, resulting in a change in the charge carriers’ concentrations [65,66,67]. In the case of p-type semiconductors like CuO, these adsorbed species are expected to lead to an increase in the resistance, due to an increment in the holes’ concentration close to the surface [65,68,69]. For this reason, all following electrical characterizations were performed in vacuum at the base pressure of 2 × 10^−3^ mbar. The electrical properties of the NW were evaluated in four-terminal configuration to avoid the effect of the contacts on electronic conductivity previously discussed by analyzing the two-terminal configuration, allowing the extrapolation of the real electrical conductivity of the single CuxO NW without any influence related to the two back-to-back Schottky diodes in series with the NW.

Figure 4c reports the comparison of the two-terminal and four-terminal configurations, in the range of ± 5 mV. As expected, in the two-terminal curve, the NW behavior was dominated by the two Schottky junctions at the Cu/Cu_x_O contacts [60]. The resistance of the NW measured in four-terminal configuration was (1.3 ± 0.3) MΩ. In this framework, the contact resistance of the two junctions was estimated as the difference between the resistance in the four-terminal and the two-terminal configurations, due to the symmetric behavior of the two Schottky diodes, which give the same contribution to the final contact resistance, following the equation $R_{2T}-R_{4T}={2R}_{C}$ [70,71]. The contact resistance was extrapolated to be 2R_C_ = (5 ± 0.6) GΩ, giving a dominant contribution to the total resistance of the device.

Assuming that the cross-section of the NW is perfectly circular, the conductivity of the Cu_x_O NW can be estimated to be (7.6 ± 0.5) × 10^−2^ S∙cm^−1^. It is worth noticing that these results highlight that the four-terminal configuration is required when the electronic properties of single nanostructures need to be assessed. Table 1 reports a comparison of the estimated NW conductivity with previous results from the literature on Cu_x_O NWs grown through thermal oxidation. The differences in these results could be attributed to many aspects including (i) the different Cu_x_O NW growth technique, (ii) the different NW structural/chemical properties, (iii) the different NW size that can influence the conducting mechanism, and (iv) the different device configurations (in the case of two-terminal configurations, the conductivity is expected to be strongly affected by the contact resistance, which also depends on the choice of metal electrodes).

### 3.5. Thermal Properties

The thermal response of the NWs was studied by means of SThM, which allowed the realization of high-resolution thermal images of the sample and the evaluation of the NWs’ thermal conductivity. An example of the topography and thermal signal image of a single NW is shown in Figure 5. Data processing consisted of obtaining a probe calibration curve from bulk and thin-film samples, which was used to determine the effective thermal conductivity of the NW, as illustrated in Figure 6. For this purpose, a profile across the SThM signal of the NW was taken, and it proved to be highly affected by topography artifacts. Therefore, the peak value corresponding to the top of the NW and the minimum probe–sample contact radius was used, similarly to the numerical model discussed below. The measured effective thermal conductivity of the NW at its top was (2.6 ± 0.5) W∙m^−1^∙K^−1^. Uncertainties were determined through calibration curve fitting and data variance.

To relate the effective thermal conductivity to thermal conductivity of the NW itself, a simple Finite Difference method (FDM) model was used [74]. The model assumed that the heat transfer was diffusive, which might be correct for the amorphous NW; however, it might have large errors for both air heat transfer, small probe–sample separations, and crystalline materials. The FDM model provides heat power flowing through the probe–sample system, which is not the same signal as SThM voltage difference. The SThM voltage difference relies on many factors due to the measurement setup and environmental conditions; therefore, converting it to heat power is not a simple task. To relate the numerical model to the experimental data, the sample modeling developed by Glasgow University was used [42]. As the effective thermal conductivities of the thin-film sample were known, this allowed us to set up a relation between the effective thermal conductivity, measured using SThM, and the heat power in the FDM model. Simulations were carried out with different conditions, to evaluate a simple estimation of the uncertainty of the numerical model, and a better numerical model is needed to prevent potential model-related biases. The estimated thermal conductivity of the NW under these assumptions is (3.1 ± 1.2) W∙m^−1^∙K^−1^. As mentioned before, the whole measurement and data interpretation procedure was performed twice, with two different SThM probes and on the same nanowires. The spatial resolution of the measurements and numerical model for the second case was lower, and the data were more scattered; nevertheless, the results were in agreement within the estimated uncertainty.

## 4. Conclusions

In summary, we reported on the comprehensive characterization of electrical and thermal conductivity of single Cu_x_O nanowires. The crystal structure, element composition, and electrical and thermal characterizations were investigated for Cu_x_O NWs grown using thermal annealing in an ambient atmosphere. The NWs’ structural and chemical compositions were determined, revealing the presence of two copper oxide phases for the NW matrix, cuprite and tenorite, and the correct oxide for the NW itself—CuO rather than Cu_2_O. Electrical characterizations were performed, revealing an increase in the Cu_x_O NW’s conduction in vacuum due to the desorption of oxygen species, with a value of 7.6 × 10^−2^ S∙cm^−1^. The effective and real thermal conductivity of the NW were estimated by means of SThM and a model based on the FDM approach, respectively, as 2.6 W × m^−1^∙K^−1^ and 3.1 W × m^−1^∙K^−1^. This work could pave the way for developing high-performance, cost-effective, and environmentally friendly optoelectronic devices based on metal-oxide NWs grown with a bottom–up approach.

## Figures and Tables

**Figure 1 nanomaterials-13-02822-f001:**
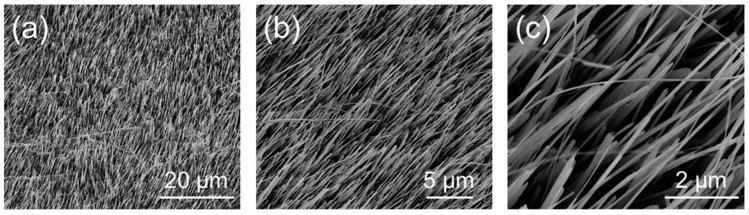
Top-view FESEM images at low- (**a**) and high- (**b**), (**c**) magnification of the Cu_x_O NW matrix after the thermal oxidation process.

**Figure 2 nanomaterials-13-02822-f002:**
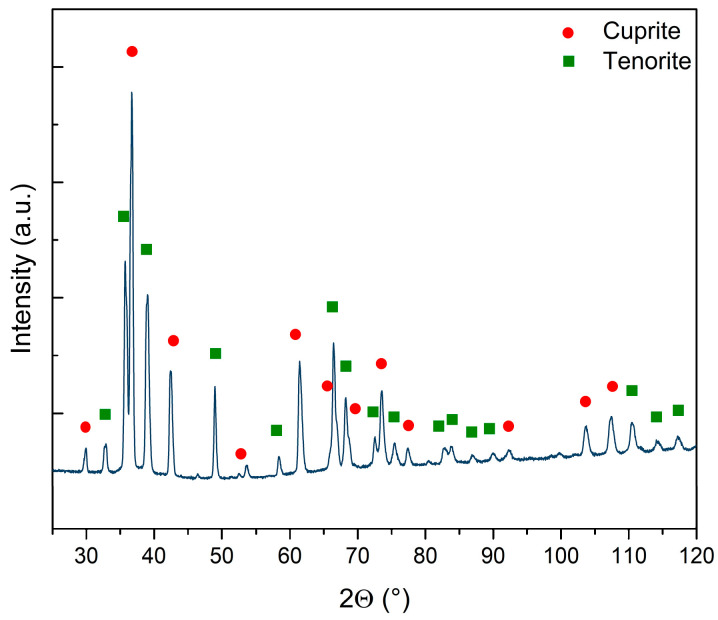
Diffractogram of the sample and qualitative assignation of the peaks to cuprite (red circles) and tenorite (green squares). The line represents the original data without background subtraction. No amorphous phases are observed.

**Figure 3 nanomaterials-13-02822-f003:**
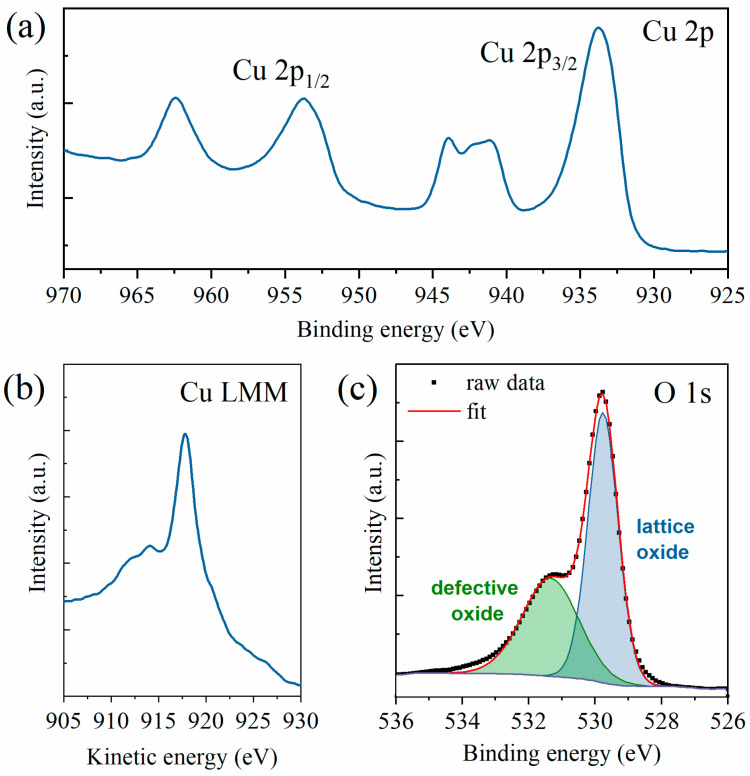
XPS HR spectra of (**a**) Cu 2p region, (**b**) Cu LMM region, and (**c**) O 1s region with peak deconvolution into lattice oxides (in blue) and defective oxides (in green).

**Figure 4 nanomaterials-13-02822-f004:**
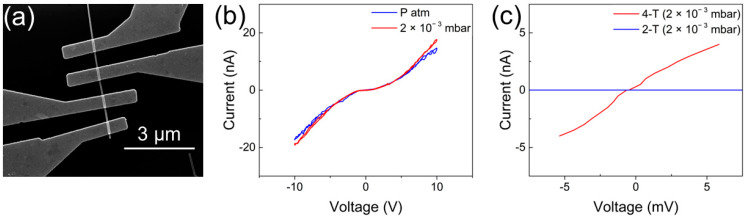
(**a**) SEM image of the single CuxO NW contacted with Cu electrodes; (**b**) 2-terminal I–V characteristics of the single CuxO NW, at ambient pressure and in vacuum; (**c**) comparison between the 4-terminal and 2-terminal *I–V* curves, highlighting the very important influence of the contact resistance on the overall device resistance.

**Figure 5 nanomaterials-13-02822-f005:**
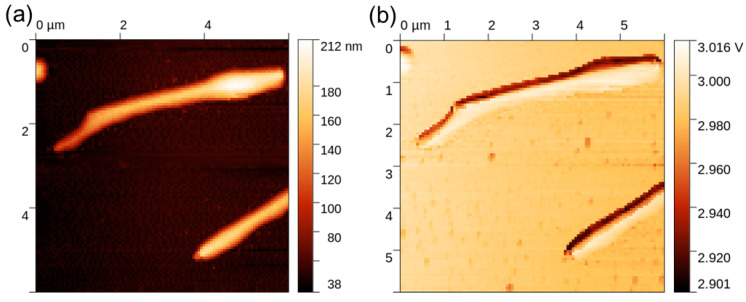
SThM measurement on an NW sample: (**a**) topography; (**b**) thermal signal.

**Figure 6 nanomaterials-13-02822-f006:**
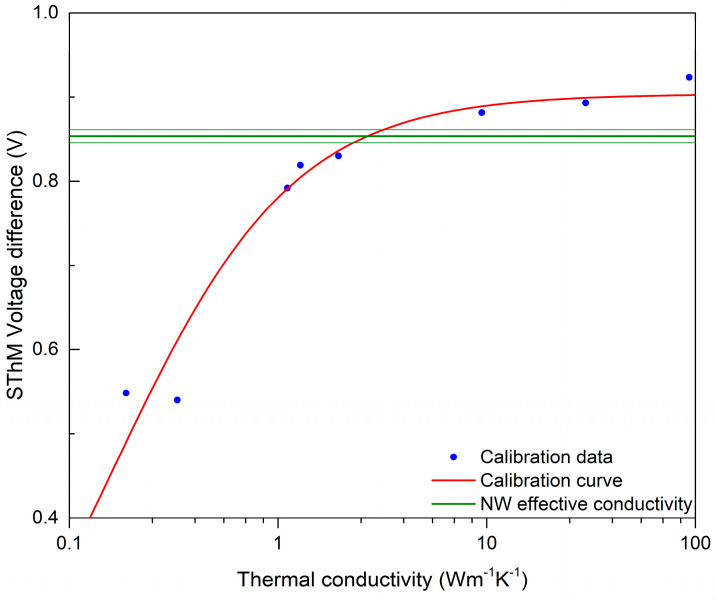
SThM calibration curve showing the intersection with measured SThM signal on top of the NW. Thin green lines show the error margins coming from the experimental data.

**Table 1 nanomaterials-13-02822-t001:** Summary of conductivities obtained in the literature, with relative electrodes, NW sizes, and electrode configurations. All the reported works are related to Cu_x_O NWs grown through thermal oxidation.

Ref	Wiring Metal	NW Size (Diameter, Length)	Electrode Configuration	σ @ 300 K (Air/Vacuum)
This work	Cu (50 nm)	64 nm, 640 nm	2T and 4T *I–Vs*	7.6 × 10^−2^ S∙cm^−1^ (vacuum)
[21]	Cr/Au (10/100 nm)	250 nm, 15 μm	2T *I–V*	2.5 × 10^−4^ S∙cm^−1^ (NA)
[72]	Au by dielectrophoresis	85 nm, 2.25 μm	2T *I–V*, space charge limited current model	~1 × 10^−2^ S∙cm^−1^ (vacuum)
[61]	Pt (300 nm) by IBID, Ti/Au (100/300 nm)	~60 nm, NA	2T *I–V* back-to-back Schottky, 4T *I–V*	~5.5 × 10^−3^ S∙cm^−1^ ~8.5 × 10^−3^ S∙cm^−1^ (vacuum)
[24]	Ni/Au (10/190 nm)	143 nm, 1.9 μm	2T and 4T *I–Vs*	~4 × 10^−3^ S∙cm^−1^ (air)
[73]	Pd/Au (3/80 nm)	36–50 nm, 80 nm–2 μm	2T *I–V*	~1–0.25 × 10^−3^ S∙cm^−1^ (air)
[18]	Ti/Au (5/50 nm)	~80 nm, NA	2T *I–V*	1.1 × 10^−3^ S∙cm^−1^ (air)

## Data Availability

Not applicable.

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
