# Peer review of "Electrical and Thermal Conductivities of Single CuxO Nanowires"

_nanomaterials, 2023, doi:10.3390/nano13212822_

Round 1
Reviewer 1 Report
The paper presents the study about electrical and thermal properties of single CuxO nanowires.
Authors present electrical and thermal properties of copper oxide NWs synthetized by thermal growth directly on copper foil. Structural characterization revealed the growth process resulted in the formation of vertically aligned NWs on the Cu growth substrate. Electrical characterization of single NW-based devices, where single NWs were contacted by Cu electrodes, revealed that NWs are characterized by a conductivity of 7.6 ∙ 10-2 S∙cm-1. An effective thermal conductivity of single CuO NWs was measured using SThM, providing value of 2.6 W∙m-1∙K-1, and using a simple Finite Difference model an estimate for thermal conductivity of nanowire itself was obtained as 3.1 W∙m-1∙K-1.
Dear author, thank you very much for interesting paper about electrical and thermal properties of CuxO nanowires. I put some comments and question.
Comments:
1. There are two times Figure 1. Please correct!!!
2. What electrical properties did you investigate? Electrical resistivity, electrical permittivity, breakdown voltage, dielectric losses? Please clarify.
3. How many thermal properties did you study? Just thermal conductivity? Please clarify.
4. General comments: I do not know what electrical properties were studied, what is mentioned in title of the paper. Authors present only voltage and current. The same comments are to thermal properties. If you studied only conductivity, please tell it in the title.
5. I think, presented material is not described. Many explanations are necessary.
Author Response
Reply to the Referees’ and Editor’s comments
We thank the Editor and the Reviewers for their comments and suggestions, that have allowed us to improve the clarity and the completeness of our manuscript. We have considered to all the points raised in their reports, by providing additional details in a revised version of the manuscript. Below, we provide a point-by-point response to each of the Referee comments and a description of the changes made.
We include in the resubmission a marked-up version of the manuscript with the changes to the text in red font, for the ease of the Referees and of the Editor. We believe that, with these changes and amendments, the manuscript is now suitable for publication in Nanomaterials.
Reply to the Editor’s comments
C1. Please also enrich the manuscript word count to 4000 words. According to our rules, this is necessary for a manuscript to have at least 4000 words in order to grant the discount voucher to the authors.
R1. We thank the Editor who gave us the opportunity to provide more details and to go deeper in the description of our work. We enriched the manuscript, reaching the word count to the number requested.
C2.Additionally, please lower the self-citation rate for your manuscript (33%),
which cannot exceed 15% of all the references.
R2. We thank the Editor, as we did not cite many important works related to this very important field, such as works related on the nanowires’ growth mechanism, or related to many applications to optoelectronics, photodetector, or photovoltaics fields. We lowered the self-citation rate of the manuscript as requested.
Reply to the Reviewer 1 comments.
The paper presents the study about electrical and thermal properties of single CuxO nanowires.
Authors present electrical and thermal properties of copper oxide NWs synthetized by thermal growth directly on copper foil. Structural characterization revealed the growth process resulted in the formation of vertically aligned NWs on the Cu growth substrate. Electrical characterization of single NW-based devices, where single NWs were contacted by Cu electrodes, revealed that NWs are characterized by a conductivity of 7.6 ∙ 10-2 S∙cm-1. An effective thermal conductivity of single CuO NWs was measured using SThM, providing value of 2.6 W∙m-1∙K-1, and using a simple Finite Difference model an estimate for thermal conductivity of nanowire itself was obtained as 3.1 W∙m-1∙K-1.
Dear author, thank you very much for interesting paper about electrical and thermal properties of CuxO nanowires. I put some comments and question.
We thank the Referee very much for his appreciation of our work. We also thank him for the comments to which we will reply in detail in the following.
Comments:
C3. There are two times Figure 1. Please correct!!!
We thank the Reviewer 1 for pointing out this aspect. We corrected the error. In addition, we corrected also the label of Figure 4 (c), which was not correct (mV and not V). The new Figure 4 is reported below for the benefit of the reviewer.
Changes in the manuscript:
Figure 4. (a) SEM image of the single CuxO NW contacted with Cu electrodes; (b) 2-terminal I-V characteristics of the single CuxO NW, at ambient pressure and in vacuum; (c) comparison between the 4-terminal and 2-terminal I-Vs, highlighting the very important influence of the contact resistance on the overall device resistance.
C4 What electrical properties did you investigate? Electrical resistivity, electrical permittivity, breakdown voltage, dielectric losses? Please clarify.
R4. We clarified through the manuscript what kind of electrical property we investigated. In order to make this aspect more clear to the reader, we have amended the title, the introduction and the materials & methods section.
Changes in the manuscript:
>>>Title: “Electrical and thermal conductivities of single CuxO nanowires”
>>>Introduction line 64: “In this paper, we report on electrical and thermal conductivities of CuxO NWs obtained by thermal oxidation of Cu foil.”
>>>Subsection 2.4 line 144: “This configuration allowed us to perform both 2-wire and 4-wire characteristics and to extrapolate the electrical conductivity of the material.”
C5. How many thermal properties did you study? Just thermal conductivity? Please clarify.
R5. In this work, we have analyzed thermal conductivity of single CuxO nanowires. Since we agree with the reviewer that this aspect was not clear for the reader, we have clarified it through the manuscript (see also reply R4)
Changes in the manuscript:
>>>Title: “Electrical and thermal conductivities of single CuxO nanowires”
>>>Introduction line 64: “In this paper, we report on electrical and thermal conductivities of CuxO NWs obtained by thermal oxidation of Cu foil.”
>>>Subsection 2.5 line 152: ” Thermal conductivity of single NWs was measured using SThM on Dimension Icon Scanning Probe Microscope (Bruker Corporation, Billerica, MA, USA) with VITA SThM electronics and VITA-DM-GLA probes (Anasys Instruments Corp, Santa Barbara, CA, USA).”
C6. General comments: I do not know what electrical properties were studied, what is mentioned in title of the paper. Authors present only voltage and current. The same comments are to thermal properties. If you studied only conductivity, please tell it in the title.
R6. We thank the Referee for suggesting better clarity for what concerns the electrical and thermal properties investigated in this manuscript. We already clarified these aspects in replies R5 and R6 to comments C5 and C6, respectively.
C7. I think, presented material is not described. Many explanations are necessary.
R7. We thank the Reviewer for suggesting to provide more details about the materials, therefore we enriched the Introduction, the description of Materials & Methods and Results. In particular, we increased the amount of details reported in Materials & Methods section.
Changes in the manuscript:
>>>Subsection 2.2 line 89: “The morphology of the samples obtained by thermal oxidation was investigated through Field Emission Scanning Electron Microscopy (FESEM), by FEI Inspect F system (Field Electron & Ion Company, Hillsboro, OR, USA), at an applied voltage of 10 kV, and a working distance of 10 mm, varying the magnification from 2000 times to 50000 times depending on the features size. ”
>>>Subsection 2.2 line 97: “XRD data were recorded at 20 °C, 4.6 bar and 20 % of humidity. The sample was placed onto a "zero background" holder thermalized at room temperature. The instrumental broadening was computed by Caglioti-equation based on the reflections of a standard LaB6 powder NIST660a. The measurements were carried out in continuous mode with a step size of 2 θ = 0.0131° and a data time per step of 1500 s. QualX equipped with COD-database was employed for the qualitative phase determination and MAUD free software for the quantitative analysis and the refinement. By recording multiple reflections at different times, no variations on the sample were observed in 1 day of experiment, suggesting stability of the NWs at such room-conditions and exposure-dose.”
>>>Subsection 2.2 line 111: “Wide-energy and high-resolution (HR) XPS spectra were collected in ultra-high vacuum (2 ∙ 10-10 mbar) after leaving samples degassing overnight. The analyses were repeated on three different areas of the sample to ensure compositional homogeneity and processed using CasaXPS software (version 2.3.18).”
>>>Subsection 2.3 line 130: “A FEI Quanta 3D (Field Electron & Ion Company, Hillsboro, OR, USA) equipped with a Nabity NPGS pattern generator (JC Nabity Lithography Systems, Bozeman, MT, USA) was employed for defining the second mask, over a PMMA A4 layer (300 nm thick), and then 50 nm of Cu was deposited through e-beam evaporation. The excess metal was finally removed by lift-off in acetone bath.”
>>>Subsection 2.4 line 138: “The electrical characterization of the device was performed through a Keithley 6430 (Keithley Instruments, Solon, OH, USA) connected to a vacuum chamber with a Leybold (Leybold GmbH, Cologne, Germany) pumping system attached to it. The single CuxO NWs were wire-bonded to a sample holder, which was previously short-circuited in order to prevent any NWs’ breakdown due to electrical discharge during the wire-bonding. The sample holder was then screwed inside of the vacuum chamber. This configuration allowed us to perform both 2-wire and 4-wire characteristics and to extrapolate the electrical conductivity of the material. At first, the characterization of the 2-terminal device was performed, to understand the junction behavior and the contact resistance: a voltage ramp was performed, first in ambient pressure and then in vacuum, at the base pressure of 2 ∙ 10-3 mbar, allowing a comparison of the electrical response depending on the ambient conditions. After that the 4-terminal characteristic of the CuxO NW was measured, to understand the NW's electrical conductivity.”
>>>Subsection 3.1 line 178: “The morphology of the NWs array composed of vertically aligned CuxO NWs obtained via the bottom-up thermal oxidation on the Cu foil was investigated by FESEM. Figure 1 reports the low (Figure 1(a)), high-magnification (Figure 1(b) - (c)) images of the CuxO NW matrix. Besides revealing the presence of CuxO NWs at the surface of the copper foil, the low-magnification image highlights the uniformity and homogeneity of the NWs growth over large scale, while the high-magnification images show the NW forest and their vertical alignment. In particular, SEM imaging revealed that CuxO NWs are characterized by a diameter of ~ 60 nm.”
>>>Subsection 3.2 line 201: “When copper foil is exposed to high temperatures in ambient atmosphere, the surface undergoes oxidation. This process first leads to the formation of a Cu2O layer, followed by the subsequent development of a layer consisting of CuO on the top of Cu2O [42]. The growth of CuO NWs relies on copper diffusion from the substrate foil through both the oxide layers, via stress induced at the interface of these layers [43–46].”
>>>Subsection 3.3 line 216: “As previously mentioned, the spectra were acquired in three different areas of the sample. However, since the spectra were almost superimposable, thus showing an excellent homogeneity, a representative spectrum is here shown and the average value for the Auger parameter calculations is reported.”
>>>Subsection 3.4 line 256: “This phenomenon could be related to the adsorbed species, which in this case led to the reduction of the NW conductivity [23,37,61,62]. When metal oxide NWs are exposed to air, oxygen species get adsorbed due to the interaction of oxygen molecules with crystal defects sites present on the NW surface, resulting in a change of charge carriers’ concentrations [63–65]. In case of p-type semiconductors like CuO, these adsorbed species are expected to lead to an increase in the resistance, due to an increment in the holes’ concentration close to the surface [64,66,67]. For this reason, all following electrical characterizations were performed in vacuum at the base pressure of 2 ∙ 10-3 mbar. Electrical properties of the NW were evaluated in 4-terminal configuration to avoid the effect of contacts on electronic conductivity previously discussed by analyzing the 2-terminal configuration, allowing the extrapolation of the real electrical conductivity of the single CuxO NW without any influence related to the two back-to-back Schottky diodes in series with the NW.”
>>>Subsection 3.4 line 271: ”In this framework, the contact resistance of the two junctions was estimated as the difference between the resistance in 4-terminal and in 2-terminal configurations, due to the symmetric behavior of the two Schottky diodes which give the same contribute to the final contact resistance, following the equation [68,69].”
>>>Subsection 3.5 line 321: “The estimated thermal conductivity of the NW under these assumptions is (3.1 ± 1.2) W∙m-1∙K-1. As mentioned before, the whole measurement and data interpretation procedure was performed twice, with two different SThM probes and on the same nanowires. The spatial resolution of the measurements and numerical model for the second case was lower and the data were more scattered, nevertheless results were in agreement within the estimated uncertainty.”
>>>Section 4 line 333: “In summary, we reported on comprehensive characterization of electrical and thermal conductivity of single CuxO nanowires. The crystal structure, element composition, electrical and thermal characterizations were investigated for CuxO NWs grown by thermal annealing in ambient atmosphere. The NWs’ structural and chemical compositions were determined, revealing the presence of two copper oxide phases for the NWs matrix, cuprite and tenorite, and the correct oxide for the NW itself, CuO. Electrical characterizations were performed, revealing an increase of the CuxO NW conduction in vacuum, with a value of 7.6 ∙ 10-2 S∙cm-1. The effective and real thermal conductivity of the NW were estimated by means of SThM and a model based on FDM approach, respectively of 2.6 W∙m-1∙K-1 and 3.1 W∙m-1∙K-1. The work reported could pave the way for developing high-performance, cost-effective, and environmentally friendly optoelectronic devices based on metal-oxide NWs grown with a bottom-up approach.”
Reply to the Reviewer 2 comments.
Dear Editor,
The authors present a study on obtaining copper oxides in the form of nanowires directly on the parent metal. The obtained results are of interest in applied microelectronics. I recommend the publication of this work although some improvements are needed.
We thank the Referee very much for his appreciation and for recommending that the paper deserves publication. We also thank him for the comments to which we will answer in detail.
Comments:
C8. In the abstract, the abbreviation for nanowires (NWs) is given, it is recommended to also give for The Scanning Thermal Microscopy (SThM), it is only given in the subsection 2.5;
R8. We thank the Reviewer 2 for his careful reading. We corrected the term as suggested in the abstract.
Changes in the manuscript:
>>>Abstract line 22: “An effective thermal conductivity of single CuO NWs placed on a substrate was measured using Scanning Thermal Microscopy (SThM), providing value of 2.6 W∙m-1∙K-1, and using a simple Finite Difference model an estimate for thermal conductivity of nanowire itself was obtained as 3.1 W∙m-1∙K-1.”
C9. Also in the introduction it is recommended to give the abbreviations for “cuprite (Cu2O) and tenorite (CuO)”;
R9. We added the abbreviations suggested in the introduction.
Changes in the manuscript:
>>>Introduction line 66: “X-ray diffraction (XRD) revealed the presence of two copper oxide phases, cuprite (Cu2O) and tenorite (CuO), and information on the NWs’ growth.”
C10. The description in the Introduction is recommended to be supported by several concrete references;
R10. We thank Reviewer who suggested to provide more references for supporting our Introduction. We have improved the introduction section by adding also new references.
C11. XRD characterizations were done in normal atmosphere (humidity RH?), XPS in vacuum, electrical characterizations were done in 10-3 mbar; it is recommended to discuss these aspects as well.
R11. We included in the manuscript more information regarding the ambient conditions at which the characterizations were done, in particular:
>>>Subsection 2.2 line 97: “XRD data were recorded at 20 °C, 4.6 bar and 20 % of humidity. The sample was placed onto a "zero background" holder thermalized at room temperature. The instrumental broadening was computed by Caglioti-equation based on the reflections of a standard LaB6 powder NIST660a. The measurements were carried out in continuous mode with a step size of 2 θ = 0.0131° and a data time per step of 1500 s. QualX equipped with COD-database was employed for the qualitative phase determination and MAUD free software for the quantitative analysis and the refinement. By recording multiple reflections at different times, no variations on the sample were observed in 1 day of experiment, suggesting stability of the NWs at such room-conditions and exposure-dose.”
>>>Subsection 2.2 line 111: “Wide-energy and high-resolution (HR) XPS spectra were collected in ultra-high vacuum (2 ∙ 10-10 mbar) after leaving samples degassing overnight. The analyses were repeated on three different areas of the sample to ensure compositional homogeneity and processed using CasaXPS software (version 2.3.18). ”
>>>Subsection 2.4 line 138: “The electrical characterization of the device was performed through a Keithley 6430 (Keithley Instruments, Solon, OH, USA) connected to a vacuum chamber with a Leybold (Leybold GmbH, Cologne, Germany) pumping system attached to it. The single CuxO NWs were wire-bonded to a sample holder, which was previously short-circuited in order to prevent any NWs’ breakdown due to electrical discharge during the wire-bonding. The sample holder was then screwed inside of the vacuum chamber. This configuration allowed us to perform both 2-wire and 4-wire characteristics and to extrapolate the electrical conductivity of the material. At first, the characterization of the 2-terminal device was performed, to understand the junction behavior and the contact resistance: a voltage ramp was performed, first in ambient pressure and then in vacuum, at the base pressure of 2 ∙ 10-3 mbar, allowing a comparison of the electrical response depending on the ambient conditions. After that the 4-terminal characteristic of the CuxO NW was measured, to understand the NW's electrical conductivity.”

Reviewer 2 Report
10.04.2023
Nanomaterials
Title: Electrical and thermal properties of single CuxO nanowires
Dear Editor,
The authors present a study on obtaining copper oxides in the form of nanowires directly on the parent metal. The obtained results are of interest in applied microelectronics. I recommend the publication of this work although some improvements are needed:
- In the abstract, the abbreviation for nanowires (NWs) is given, it is recommended to also give for The Scanning Thermal Microscopy (SThM), it is only given in the subsection 2.5;
- Also in the introduction it is recommended to give the abbreviations for “cuprite (Cu2O) and tenorite (CuO)”;
- The description in the Introduction is recommended to be supported by several concrete references:
- XRD characterizations were done in normal atmosphere (humidity RH?), XPS in vacuum, electrical characterizations were done in 10-3 mbar; it is recommended to discuss these aspects as well...
Best regards,

see the attachment
Author Response

(The authors gave the same response as above.)

Round 2
Reviewer 1 Report
All my comments are included by the authors. I think, the paper is ready to be published.